

# High glucose treatment promotes extracellular matrix proteome remodeling in Müller glial cells

Sandra Sagmeister[1,2], Juliane Merl-Pham[2], Agnese Petrera[2], Cornelia A. Deeg[1] and Stefanie M. Hauck[2]

[1] Chair of Physiology, Department of Veterinary Sciences, LMU Munich, Martinsried, Germany
[2] Research Unit Protein Science and Metabolomics and Proteomics Core Facility, Helmholtz Center Munich, German Research Center for Environmental Health GmbH, Munich, Germany

## ABSTRACT

**Background**. The underlying pathomechanisms in diabetic retinopathy (DR) remain incompletely understood. The aim of this study was to add to the current knowledge about the particular role of retinal Müller glial cells (RMG) in the initial processes of DR.

**Methods**. Applying a quantitative proteomic workflow, we investigated changes of primary porcine RMG under short term high glucose treatment as well as glycolysis inhibition treatment.

**Results**. We revealed significant changes in RMG proteome primarily in proteins building the extracellular matrix (ECM) indicating fundamental remodeling processes of ECM as novel rapid response to high glucose treatment. Among others, Osteopontin (SPP1) as well as its interacting integrins were significantly downregulated and organotypic retinal explant culture confirmed the selective loss of SPP1 in RMG upon treatment. Since SPP1 in the retina has been described neuroprotective for photoreceptors and functions against experimentally induced cell swelling, it's rapid loss under diabetic conditions may point to a direct involvement of RMG to the early neurodegenerative processes driving DR. Data are available via ProteomeXchange with identifier PXD015879.

Corresponding authors
Cornelia A. Deeg,
Cornelia.Deeg@lmu.de,
deeg@tiph.vetmed.uni-muenchen.de
Stefanie M. Hauck,
hauck@helmholtz-muenchen.de

## INTRODUCTION

The worldwide prevalence for diabetes in adults is increasing and is predicted to exceed ten percent of world population by 2040 (*Ogurtsova et al., 2017*). Diabetic retinopathy (DR) is a severe and frequent comorbidity of diabetes and a leading cause of vision loss (*Ducos et al., 2020*; *Graue-Hernandez et al., 2020*; *Leasher et al., 2016*). However, the underlying pathomechanisms remain to be comprehensively elucidated (*Mesquida, Drawnel & Fauser, 2019*). DR was long proposed to be mainly a microvascular disease, but is today perceived rather as a neurodegenerative disorder in its early stages (*Kadlubowska et al., 2016*). Neurodegeneration occurs even before any vascular alterations can be detected and cannot be reversed (*Kadlubowska et al., 2016*). Thus, there is an urgent need to learn about initial
pathomechanisms involved in the early neurodegenerative process. Retinal Müller glial cells (RMG) are a major source for neuroprotective signals (*Del Rio et al., 2011*; *Hauck et al., 2008*; *Hauck, Toerne & Ueffing, 2014*) also in the diabetic mouse retina (*Fu et al., 2015*). Thus, the novel concept of modulating RMG to improve or re-establish retinal function is gaining increasing attention. A detailed understanding of RMG participation in the onset of neurodegeneration could guide novel therapeutic intervention strategies and, in best case, help prevent neurodegeneration.

The retina is composed of a complex network of highly specialized cells (*Mazade & Eggers, 2020*). RMG are the dominant macroglial cell type within the retina and retinal health depends on their undisturbed functions (*Bringmann et al., 2006*). RMG seem to be fundamentally involved in pathologic mechanism of DR (*Coughlin, Feenstra & Mohr, 2017*). They mediate the local inflammatory response, trigger pathological neovascularization and fibrosis, participate in damage of blood retinal barrier and downregulate potassium channels vital to retinal fluid homeostasis and synaptic function (*Coughlin, Feenstra & Mohr, 2017*; *Sorrentino et al., 2016*; *Subirada et al., 2018*). However, little is known about the exact mechanism in RMG and changes on protein level have not yet been studied in DR (*McDowell et al., 2018*).

Due to the high anatomical and functional similarities of pig and human eyes, pig ocular tissue has proven a promising model for translational research (*Giese et al., 2020*; *Kleinwort et al., 2017*; *Menduni et al., 2018*; *Renner et al., 2020*). While other animal models fall short in mimicking the clinical background of DR in man (*Mi et al., 2014*), the diabetic INS$^{C94Y}$ transgenic pig is the first model to represent a broad variety of the human retinal pathology (*Kleinwort et al., 2017*; *Weigand et al., 2020*). We hence established a cell culture model for DR on isolated porcine RMG by applying diabetic conditions in vitro. Hyperglycemia is considered as major trigger of pathological events in DR (*Mathebula, 2018*) and streptozotocin-induced diabetes is known to increase retinal glucose levels in diabetic rats (*Ola et al., 2006*; *Zhang et al., 2003*). We, therefore, applied short term high glucose treatment to simulate early hyperglycemic conditions in diabetes. To meet another form of disturbed glucose metabolism, we also applied 2-deoxyglucose treatment to simulate glycolysis inhibition (*Valdes et al., 2016*). There are indications that decreased glycolysis or inactivity of glycolytic enzymes are another critical matter in the development of DR in diabetic rats (*Kanwar & Kowluru, 2009*; *Ola et al., 2006*; *Thomas et al., 2019*).

In this study, we explored proteomic changes of porcine RMG in a diabetic cell culture model and contribute to understanding RMG mediated pathomechanisms in DR. Studying RMG responses to short term high glucose and glycolysis inhibition treatment on protein level allowed us to assess the initial involvement of RMG in early stages of DR. The proteomic data was confirmed by immunohistochemical staining on porcine retinal organotypic explant cultures.

## MATERIALS & METHODS

### Preparation of primary RMG

No experimental animals were used in this study, eyes from healthy adult pigs were received fresh from a local abattoir. The eyes were removed from the animals within five minutes
after death and kept on ice in $CO_2$-independent medium (Thermo Fisher Scientific, Ulm, Germany) until preparation started within one hour. Collection and use of porcine eyes from the abattoir was approved for purposes of scientific research by the appropriate board of the veterinary inspection office Munich, Germany (permit number: DE 09 162 0008-21). Eight neuroretinae from eight animals were prepared and primary RMG were isolated as previously described (*Hauck, Suppmann & Ueffing, 2003*). Briefly, major blood vessels were removed, each retina was mechanically cut and retinal pieces were washed twice in Ringer's solution (Millipore Sigma, Darmstadt, Germany). Dissociation was obtained by treating each retina with 2.2 U of activated papain (Worthington Biochemical, Troisdorf, Germany) for 12 min at 37 °C. Papain enzyme activity was stopped by the addition of DMEM Glutamax (Thermo Fisher Scientific) with 10% fetal calf serum (Thermo Fisher Scientific). Then, 160 Kunitz-units of DNase (Millipore Sigma, Darmstadt, Germany) was added and the tissue was further dissociated by gentle trituration using a fire-polished Pasteur pipette. Dissociated cells were pooled, resuspended and plated in six well format (cells from about two retinae per six well plate) in DMEM Glutamax containing 5.6 mM glucose in addition of 10% fetal calf serum and 1% Penicillin-Streptomycin (Thermo Fisher Scientific). After 24 h at 37 °C in 5% $CO_2$, medium was exchanged and debris, nonattached or loosely attached cells were removed by agitation (panning). Cells were cultivated for two weeks without passaging and DMEM Glutamax culture medium was exchanged every 48 to 72 h. Under these treatment conditions, porcine RMG cells are rapidly dominating the culture and reach purity and 80% confluence after two weeks in culture (*Hauck, Suppmann & Ueffing, 2003*).

The time point after which confluence is reached strongly depends on the total number of seeded cells. We spread the cells derived from two porcine eyes onto one 6well culture dish and reproducibly reach 80% confluence after 14 days (100% confluence is never reached in these cultures). We chose this time point for our studies because this was the earliest consistent time point with reasonably high cell yields for proteomic workflows. However, growth of RMG in vitro goes along with trans-differentiation during culturing concomitant with strong downregulation of prototype RMG marker proteins, like glutaminsynthetase (GS) (*Hauck, Suppmann & Ueffing, 2003*). The remaining expression of GS after only 7 days in culture (*Hauck, Suppmann & Ueffing, 2003*) is mainly due to remaining resting cells which have not yet started growing (and simultaneously loosing RMG markers). Since the mixture of non-growing and growing, yet de-differentiated cells is a rather unstable experimental condition, we chose the time point 14 days, where the cells appear homogeneously fibroblast-like, even though they have undergone trans-differentiation.

Cells were imaged by differential interference contrast on a Leica DMi8 microscope with the HC PL Fluotar L 20x/0.40 DRY objective lens using a Leica DFC365FX-722433014 camera and processed by the Leica Application Suite LASX (version 3.03, Leica, Wetzlar, Germany) (Fig. 1).

## Diabetic treatment of RMG

Experiments were performed after 14 days in culture when cells reached a confluence of approximately 80%. Diabetic treatment was performed for 72 h, with replacement of the

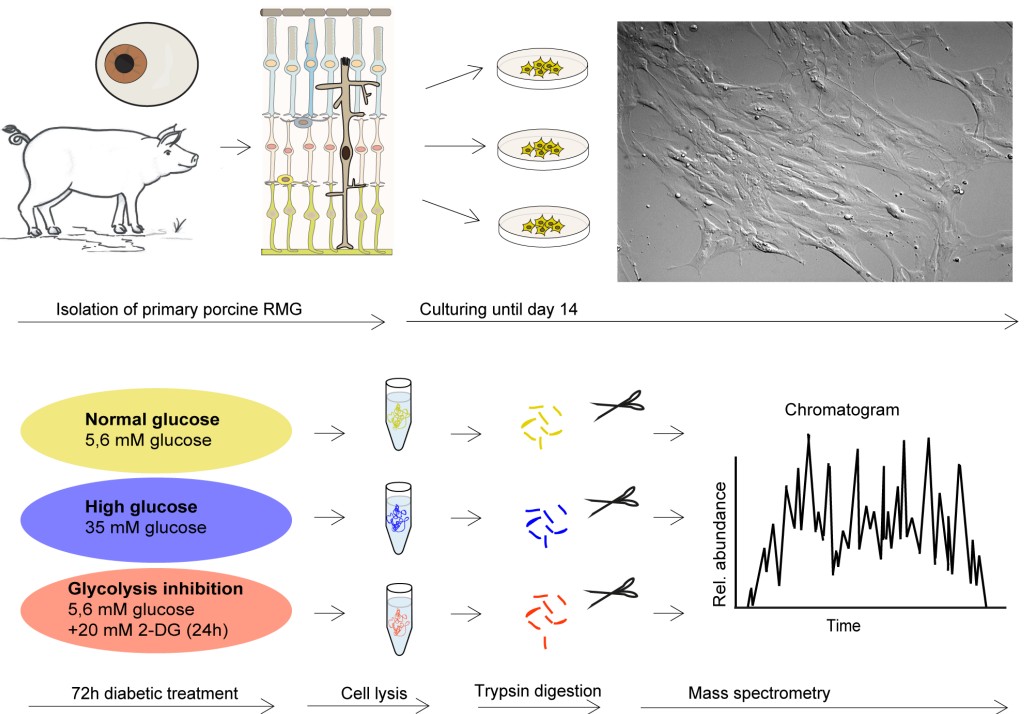

**Figure 1** **Retinal RMG proteome profiling in a diabetic cell culture model system.** Experimental workflow: Primary retinal RMG were isolated from eyes of adult pigs. After 14 days in vitro, cells were treated in three different conditions for 72 h. After treatments, cells were lysed and proteolysed proteins quantified by label-free LC-MS/MS-based proteomic analysis applying a data-independent acquisition mode.

respective culture medium every 24 h. The cells under normoglycemic conditions ($n = 6$ replicates, represented by different culture wells) were incubated at physiologic glucose level of 5.6 mM resembling the physiologic blood glucose level in human and porcine species (*Renner et al., 2013*). The cells under high glucose treatment conditions aiming to simulate hyperglycemic conditions ($n = 6$ replicates) were cultured with 35 mM glucose representing a severe hyperglycemic state in diabetic patients (*Yu et al., 2008*; *Zhu et al., 2015*) by addition of sterile filtered D-glucose (Millipore Sigma, Darmstadt, Germany). The cells under glycolysis inhibition ($n = 6$ replicates) were cultured under normoglycemic conditions with addition of 20 mM 2-deoxy-D-glucose (2-DG, Millipore Sigma, Darmstadt, Germany) during the last 24 of the 72 h. 2-DG is a synthetic glucose analog that is assimilated and subsequently phosphorylated by mammalian RMG in vitro (*Poitry-Yamate & Tsacopoulos, 1991*; *Toft-Kehler et al., 2017*). Uptake of 2-DG in mammalian (cancer) cells competes with glucose, as it is transferred by glucose transporters (GLUTs) (*Zhang et al., 2016*). Phosphorylated 2-DG cannot be further metabolized and accumulates leading to non-competitive inhibition of hexokinase, thus disrupting glycolysis. In mouse retinae, a concentration of 1 mM 2-DG in the absence of glucose is sufficient to almost completely disrupt energy production ex vivo within 45 min (*Chertov et al., 2011*). In the presence of glucose, 2-DG was applied in a ratio of 1:1 competing glucose to achieve glycolysis inhibition within 24 h (*Valdes et al., 2016*). In this experiment we applied an about 3.5

times excess of 2-DG compared to the glucose level to achieve glycolysis inhibition (20 mM 2-DG and 5.6 mM glucose). Wells representing the different treatments were randomized across three six well plates. Viability of cells at the end of the experiment was examined under microscope and estimated over 80%.

## Sample preparation for mass spectrometry

After treatment, cells were washed trice in ice cold phosphate buffered saline (PBS). Plates were put on ice and 100 µl lysis buffer (PBS with 1% NP40, Roche Diagnostics, Mannheim, Germany) was added per well. After two minutes of incubation, cells from each well were scraped and collected separately. Lysis was performed by shear force and ultrasonication followed by centrifugation for ten minutes at 10.000 rcf. Protein concentration was determined by Bradford assay and equal total protein amounts (10 µg) per replicate were digested with a modified FASP procedure (*Grosche et al., 2016*; *Wisniewski et al., 2009*). Briefly, protein lysates were diluted with ammonium bicarbonate buffer (Millipore Sigma, Darmstadt, Germany) to a final volume of 400 µl, followed by reduction using 1 µl of 1 M dithiothreitol (Millipore Sigma, Darmstadt, Germany) for 30 min at 60 °C. After cooling down to room temperature, 8 M urea buffer pH 8.5 (Millipore Sigma, Massachusetts, USA) was added to a final volume of 1 ml and proteins were carbamidomethlylated with 10 µl of 300 mM iodoacetamide (Millipore Sigma, Darmstadt, Germany) for 30 min at room temperature in the dark. Two µl of 1 M dithiothreitol was added to quench unreacted iodoacetamide, and protein lysates passed 30 kDa centrifugal filters, Vivacon 500 (Sartorius, Göttingen, Germany). After washing three times with 200 µl of urea buffer and three times with 100 µl of 50 mM ammonium bicarbonate buffer, the proteins on the filters were subjected to a two hours digest at room temperature with 0.5 g of Lysyl Endopeptidase (Wako, Osaka, Japan) followed by tryptic digest (1g of trypsin, Promega, Madison, USA) over-night at 37 °C. Peptides were collected by centrifugation through the filter and acidified with trifluoroacetic acid (Applied Biosystems, Foster City, USA) to a final pH 2.

## LC-MS/MS analysis

Approximately 0.5 µg of peptides per sample were measured in a randomized fashion on a Q-Exactive HF mass spectrometer online coupled to an Ultimate 3000 nano-RSLC (Thermo Fisher Scientific, Ulm, Germany) in data-independent acquisition (DIA) mode as previously described (*Lepper et al., 2018*; *Mattugini et al., 2018*). Briefly, peptides were loaded automatically on a trap column (300 µm inner diameter ×5 mm, Acclaim PepMap100 C18, 5 µm, 100 Å; LC Packings, Sunnyvale, USA) prior to C18 reversed phase chromatography on the analytical column (nanoEase MZ HSS T3 Column, 100 Å, 1.8 µm, 75 µm × 250 mm; Waters, Milford, USA) at 250 nl/min flow rate in a 105 min non-linear acetonitrile gradient from 3 to 40% in 0.1% formic acid. Profile precursor spectra from 300 to 1,650 m/z were recorded at 120,000 resolution and a maximum injection time of 120 ms for achieving an *automatic gain control* (*AGC*) value of *3e6*. Subsequently fragment spectra were recorded in 37 overlapping DIA isolation windows (1 Da overlap on each side) of variable size covering in total 300 to 1650 m/z, each at 30,000 resolution with an AGC target of 3e6 and a normalized collision energy of 27.

## Quantitative analysis

The recorded raw files were analyzed using the Spectronaut software (version 12, Biognosys, Schlieren, Switzerland (*Bruderer et al., 2015*; *Bruderer et al., 2016*) with a peptide identification false discovery rate setting of <1%, using an in-house pig spectral library which was generated using Biognosys Spectronaut 12 and the Ensembl Pig database (release 75, Sscrofa10.2). Quantification was based on MS2 area levels of all unique proteotypic peptides per protein fulfilling the percentile 0.2 setting. Normalized protein quantifications were exported and used for calculations of fold-changes and significance values. Each group originally included six replicates; in the normoglycemic control group two replicates were excluded due to a deviating abundance response that could not be solved by normalization.

## Statistics, graphics, principal component analysis, pathway enrichment analysis, volcano plots, Venn diagram and network generation

To analyze the differential protein abundance between the diabetic treatment groups and the control group, a two-tailed, unpaired student's $t$-test was performed on log2 transformed normalized abundances. Proteins were considered significantly changed upon the following criteria: (a) quantified with $\geq 2$ unique peptides and (b) $t$-test $p$-value <0.05. Additionally, a fold change cut-off of $\geq 1.33$-fold or $\leq 0.75$-fold was applied. To justify the chosen fold change cut-off, we calculated the average variation coefficient CV of the individual groups (resulting in average CVs of 13.2% in the normoglycemic, 12.0% in the high glucose and 14.1% in the glycolysis inhibition group) and chose the fold change cut-off to exceed twice the maximal group CV.

Pathway enrichment analysis (Fig. 2) was performed separately for the high glucose and the glycolysis inhibition treatment group with Genomatix Generanker (Genomatix Software GmbH, Munich, Germany, https://www.genomatix.de/) using EIDorado (version 04-2019) and Literature Mining Database (version 02-2019). Inclusion criteria for proteins were (a) $p$-value <0.05, (b) quantification with $\geq 2$ unique peptides and (c) $a \geq 1.33$ fold or $\leq 0.75$-fold differential abundance level. Gene names of input proteins were uploaded using H. sapiens as the organism for background list.

To illustrate overlap of proteins from the high glucose and the glycolysis inhibition treatment groups (Fig. 3A) that fulfil criteria (a) $p$-value <0.05 and (b) quantification with $\geq 2$ unique peptides from high glucose and glycolysis inhibition group and (c) $a \geq 1.33$-fold or $\leq 0.75$-fold differential abundance level in these both groups in comparison to the normoglycemic group, we employed BioVenn (Tim Hulsen, http://www.biovenn.nl) (*Hulsen, De Vlieg & Alkema, 2008*).

For generation of the network of proteins (Fig. 3B), we used STRING (Search Tool for the Retrieval of Interacting Genes/ Proteins, version 11.0, STRING consortium 2017, https://string-db.org) which belongs to the ELIXIR's Core Data Resources (*Szklarczyk et al., 2019*). STRING was employed using "highest confidence" regarding the interaction score, with a score limit of 0.9 describing the approximate probability that a predicted link exists between two proteins in the same metabolic map in the KEGG database (release 90.1). Additionally, disconnected nodes were hidden while the thickness of the connection lines

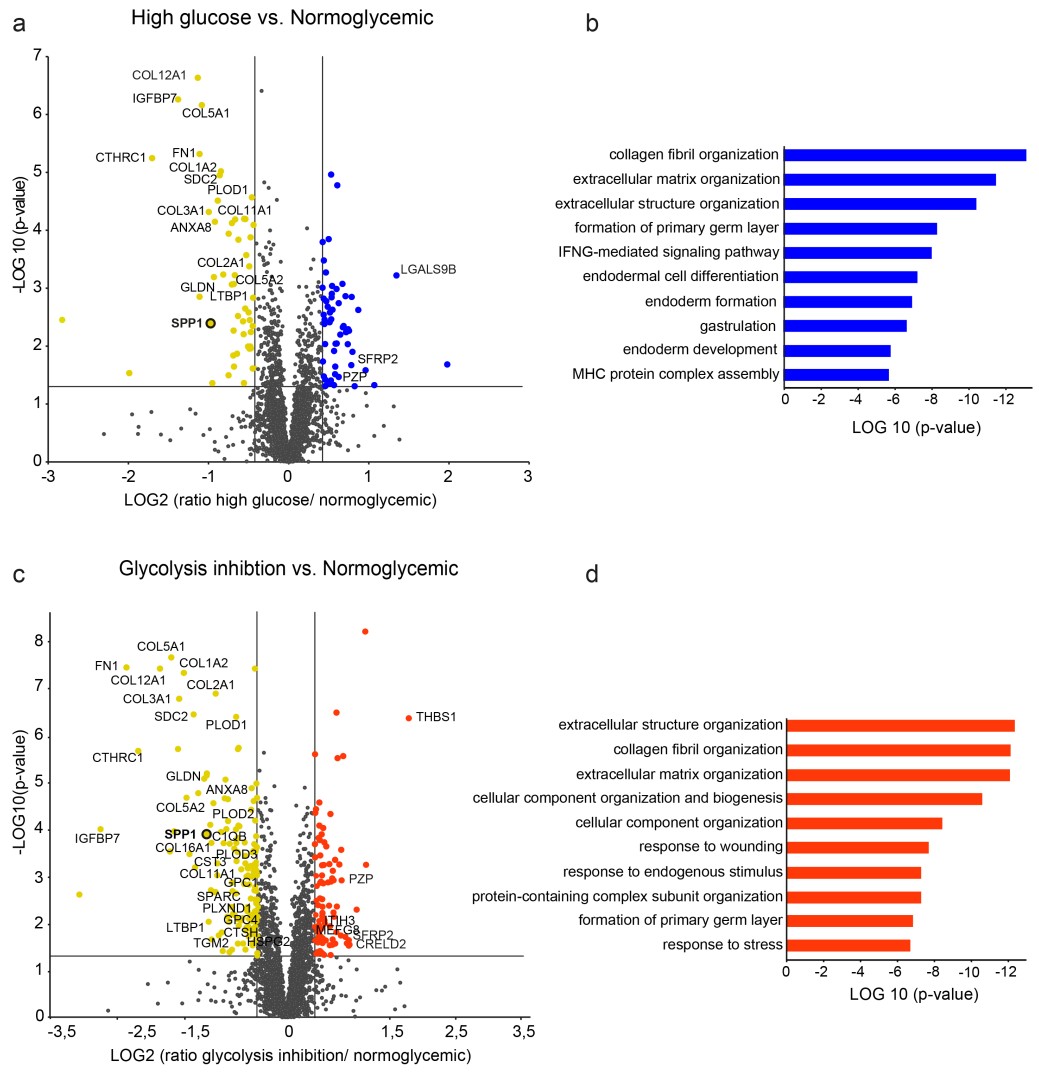

**Figure 2** **Differential protein abundances from primary porcine RMG in high glucose (A, B) or glycolysis inhibition (C, D) treatment compared to normoglycemic treatment group.** Volcano plots illustrating the log2 transformed abundance fold change of quantified proteins ($\geq$ 2 unique peptides) between the high glucose (A) or glycolysis inhibition (B) and normoglycemic group plotted against the corresponding negative log10 transformed $p$-value. Reference lines illustrate a $p$-value of 0.05 and a 1.33-fold or 0.75-fold differential abundance level. Significantly ($p$-value < 0.05) changed proteins with $\geq$ 1.33 (blue/red) or $\leq$ 0.75-fold (yellow) differential expression levels are colored. ECM-related proteins are labeled by gene names. (B), (D) Top ten most significantly altered biological processes in hyperglycemic (B) or glycolysis inhibition (D) treatment group according to Genomatix Gene Ranker Database. The three highest ranking processes in each condition belong to the context of ECM.

indicates the strength of the data support. All of the presented connection lines reaching the maximum thickness indicated the maximal strength of the data support. As input format we chose gene names and H. sapiens as organism.

## Porcine neuroretinal organotypic explant cultures

Healthy adult porcine eyes were provided by a local abattoir, removed from the animals within five minutes after death and kept on ice in $CO_2$-independent medium until preparation was completed within three hours. One explant was dissected per eye, and all of the eyes were obtained from different animals. Neuroretinal explants were dissected as described (*Taylor et al., 2014*) with some modifications. The opened eyecup was filled with prewarmed $CO_2$-independent medium and the neuroretina was carefully removed from retinal pigment epithelium by very softly flushing medium between the two structures. To ensure correct orientation concerning retinal topography, neuroretina was properly teased back into the original position with a forceps. We carefully excised circular tissue pieces of six mm diameter with a biopsy punch (PFM Medical, Köln, Germany) from the dorso-nasal position of the optic disc (Fig. 4).

Free floating neuroretinal pieces were carefully collected with the upper (large) end of a pasteur pipette connected to a pasteur pipette bulb at the broken lower end and positioned onto 0.4 µm culture plate inserts (Thermo Fisher Scientific, Roskilde, Denmark) with the inner limiting membrane side facing the insert membrane and thus providing inner retinal support (*Taylor et al., 2014*). Inserts were placed in six well plates filled with 1.5 ml RPMI 1640 medium (Thermo Fisher Scientific, Ulm, Germany) containing 11 mM glucose and supplemented with 10% fetal calf serum and 0.2% Penicillin-Streptomycin and respective treatment additives. A drop of medium was placed at the air interface on top of each explant and explants were incubated at 37 °C in 5% $CO_2$.

## Treatment of explants

Treatment started immediately after dissection. During the 72 h of treatment, culture medium in the chamber below the explant as well as the drop placed on top of the explant was replaced every 24 h. The normoglycemic group ($n = 4$ biological replicates, represented by explants from different eyes and animals) was maintained with a glucose level of 11 mM as included in RPMI 1640 medium. The high glucose group ($n = 4$ biological replicates) was cultured under conditions containing 70 mM glucose by addition of sterile filtered D-glucose. This concentration was defined in consideration of the factor of 6.25 between normoglycemic and high glucose group that was employed in the isolated RMG cell experiment before (5.6 mM normoglycemic, 35 mM high glucose, 6.25 fold increase). To investigate an osmotic effect independent of the metabolic effects of glucose (*Yu, Liu & Zhong, 2017*), we included an osmotic control group ($n = 4$ biological replicates) containing 11 mM of glucose and 59 mM D-mannose (Millipore Sigma, Darmstadt, Germany) to achieve the same osmolarity like in the high glucose group. Explants representing the glycolysis inhibition group ($n = 4$ biological replicates) were cultured with levels of 11 mM glucose with addition of 40 mM 2-DG within the last 24 of the 72 h. This concentration was determined to achieve an about 3.5-fold overdose of 2-DG compared to glucose similar to the RMG experiment (which was 20 mM 2-DG with 5.6 mM glucose).

## Immunohistochemistry of explants

After 72 h of treatment, explants were fixated, dehydrated and embedded for cryo sectioning adapted from a protocol provided by Linnea Taylor, Ophthalmology Department, Medical

Faculty, Lund University (*Taylor et al., 2014*). Briefly, prechilled (4 °C) paraformaldehyde (PFA) 4% (BosterBio, Pleasanton, USA) was drop-wise added to explants still adherent to the insert membrane over a time period of minimal five minutes until they were completely covered with PFA and fixation continued for three hours at 4 °C. PFA was removed and explants were washed three times with prechilled phosphate buffer (PB; 0.1 M, pH 7.4). Then each explant was excised with eight mm biopsy punch (PFM Medical, Köln, Germany) and transferred with the membrane to flat biopsy capsules (Cell Path, Newton, UK). Cryopreservation was performed by incubating in PB containing 15% sucrose (Millipore Sigma, Darmstadt, Germany) for four hours followed by incubation in PB containing 30% sucrose overnight at ambient temperature. Explants were taken from the capsule, moved to 12 mm ×eight mm embedding molds (Polyscienes, Niles, USA), cryo-embedded together with the membrane in OCT medium (Thermo Fisher Scientific, Ulm, Germany) and stored at −80 °C. Cryo-sections were collected from the center parts of each explant at 18 μm thickness, mounted on coated slides (Superfrost Plus, Thermo Scientific, Ulm, Germany) and stored at −20 °C.

Slides were defrosted at room temperature for 20 min and washed with PB trice for ten minutes each. To avoid unspecific antibody binding, sections were blocked in Tris-buffered saline containing 0.1% Tween 20 (Serva, Heidelberg, Germany, TBS-T), 1% bovine serum albumin (Biomol, Hamburg, Germany) and 5% goat serum (Abcam, Berlin, Germany) for one hour at ambient temperature. Sections were then co-incubated overnight at 4 °C with rabbit anti-SPP1 (1:250, OriGene, Herford, Germany) and mouse anti-vimentin (1:50, Millipore Sigma, Darmstadt, Germany) diluted in TBS-T with 1% bovine serum albumin and 5% goat serum, followed by co-incubation with goat anti-rabbit IgG AlexaFluor 488 and goat anti-mouse IgG AlexaFluor 568 (each 1:500, Thermo Fisher Scientific, Ulm, Germany) for 90 min at ambient temperature. Between primary and secondary resp. after secondary antibody incubations sections were washed three times each with PB for 10 min. After counterstaining with Hoechst (1:10,000, Thermo Fisher Scientific, Ulm, Germany) for 10 min at ambient temperature, coverslips were mounted in aqueous FluorSave cover medium (Thermo Fisher Scientific, Ulm, Germany) and photographed on a Leica DMi8 microscope with the HC PL APO 40x/0.95 DRY objective lens. Filter cubes for GFP, Texas Red and DAPI detections were used (JH Technologies). All images were captured using a Leica DFC365 FX camera, and constant settings for gain and exposure time were maintained for all sections within an experimental setup. Images were processed by the Leica Application Suite LASX (version 3.03, Leica). As control, three sections per group from different explants were stained under equal conditions with anti-rabbit-IgG (1:1,000, Abcam, Berlin, Germany) instead of anti-SPP1; no unspecific labeling was observed.

# RESULTS

## Overrepresentation of ECM-associated biological processes in RMG under diabetic culture conditions

We isolated primary porcine RMG from pig eyes, applied the diabetes short term treatment and investigated proteome-wide changes applying a mass spectrometric work flow (Fig. 1).

A total of 2,744 proteins quantified with ≥ 2 unique peptides were detected across in all treatment groups and replicates (Table S1).

In the high glucose treatment group, a total of 464 proteins were significantly differently abundant compared to the normoglycemic control group (Fig. 2A, protein-dots above the reference line for *p*-value cut-off, Table S2), thereof 103 proteins met the additional criterion of fold change cut-off (Fig. 2A, colored protein-dots beyond both reference lines for fold change cut-off). Pathway enrichment analysis of these 103 proteins revealed the top ten biological processes that were significantly enriched with a functional relationship to the input proteins (Fig. 2B). The three highest ranking pathways are strongly associated with extracellular matrix function, namely "collagen fibril organization", "extracellular matrix organization" and "extracellular structure organization" (Fig. 2B, Table S3). Accordingly, among the changed proteins were 19 members of the ECM (Table S2, according to the human matrisome database, http://matrisomeproject.mit.edu (*Naba et al., 2016*); Fig. 2A, labeled with gene symbols) of which the majority (16) showed lower abundance levels in the high glucose treatment group compared to the normoglycemic control group.

In the glycolysis inhibition treatment group, altogether 800 proteins exhibited significant changes in abundance levels compared to the normoglycemic control group (Fig. 2C, protein-dots above the reference line for *p*-value cut-off, Table S4), of which 240 proteins additionally met the fold change cut-off (Fig. 2C, colored protein-dots beyond both reference lines for fold change cut-off). Again, pathway enrichment analysis of these 240 proteins pointed to an overrepresentation of proteins belonging to the context of extracellular matrix, which was demonstrated by the three most significantly overrepresented biological processes (Fig. 2D, namely "extracellular structure organization", "collagen fibril organization" and "extracellular matrix organization"; Table S5). Correspondingly, there were 34 ECM members among the input proteins (Table S4, according to the human matrisome database; Fig. 2C labeled with gene symbols), with the majority (28) once again showing lower abundance levels in the glycolysis inhibition treatment group compared to the normoglycemic control group. Interestingly, secreted phosphoprotein 1 (SPP1) appeared in both diabetes-like conditions as one of the top changed candidates.

## Downregulation of ECM organization and cell adhesion processes under diabetic culture conditions

Both diabetic treatments caused a similar pattern of highly overrepresented processes when compared to the normoglycemic control. Hence, we aimed to clarify if this could be linked to a similar alteration in abundance level of particular proteins or groups of proteins. There were differences in the proteins with changed abundance between diabetic conditions, but 65 of all differently abundant proteins overlapped (Fig. 3A). We concluded that these 65 proteins have fundamental roles in RMG metabolism, since they reacted consistently to different changes applied to glucose metabolism by our experimental treatments. We next performed a protein enrichment analyses with STRING, applying strict evaluation criteria to investigate the major functions of these 65 proteins. These analyses confirmed enriched interactions of proteins of the biological process "ECM organization" and additionally

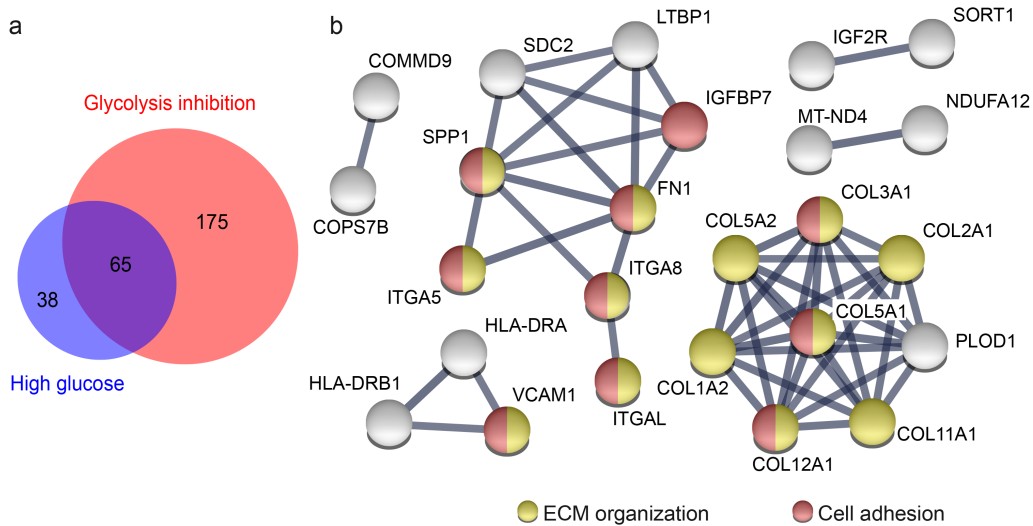

**Figure 3** **Overlap of proteins from primary porcine RMG with differential abundance in high glucose and glycolysis inhibition compared to normoglycemic treatment group points to affected ECM organization and cellular adhesions.** Overlap of proteins with differential abundance in high glucose and glycolysis inhibition compared to normoglycemic treatment group points to affected ECM organization and cellular adhesions (A) Venn diagram of all proteins with differential abundance (criteria: ≥ 2 unique peptides, $p$-value < 0.05, fold change ≥ 1.33 or ≤ 0.75-fold) in hyperglycemic (blue) and glycolysis inhibition (red) compared to normoglycemic treatment group. (B) STRING protein functional association network of 65 overlapping proteins that display differential abundance in hyperglycemic and glycolysis inhibition compared to normoglycemic treatment group. Colored protein symbols belong to the GO term biological process "ECM organization" (yellow) and "cellular adhesion" (red).

defined the biological process "cell adhesion" (Fig. 3B, according to the Gene Ontology term of biological processes). Sixty-two of 65 proteins changed their abundance in the same way, so either decreasing (39 proteins) or increasing (23 proteins) their abundance levels in both treatment groups compared to the control group. All of the proteins belonging to the "ECM organization" and "cell adhesion" biological process in the STRING network are proteins with decreased abundance in both treatment groups compared to the normoglycemic control group. That points to a downregulation of both processes in high glucose and glycolysis inhibition treatment.

Some of the candidates have functions in ECM organization as well as in cell adhesion. Among them is SPP1 (Fig. 3B), that interacts in both pathways with collagen-$\alpha$-chains, fibronectin and the alpha integrins ITGA5, ITGA8, ITGAL (Fig. 3B, Tables S2 and Table S4). We therefore decided to further investigate SPP1 in our hyperglycemia model.

## RMG-associated SPP1 expression pattern selectively diminished in retinal explants under diabetic culture conditions

SPP1 was consistently decreased in both diabetic conditions and belongs to several overrepresented biological processes in the dataset. Earlier, SPP1 was shown to play a key role in RMG mediated pathogenesis in an inflammatory eye disease (*Deeg et al., 2011*).

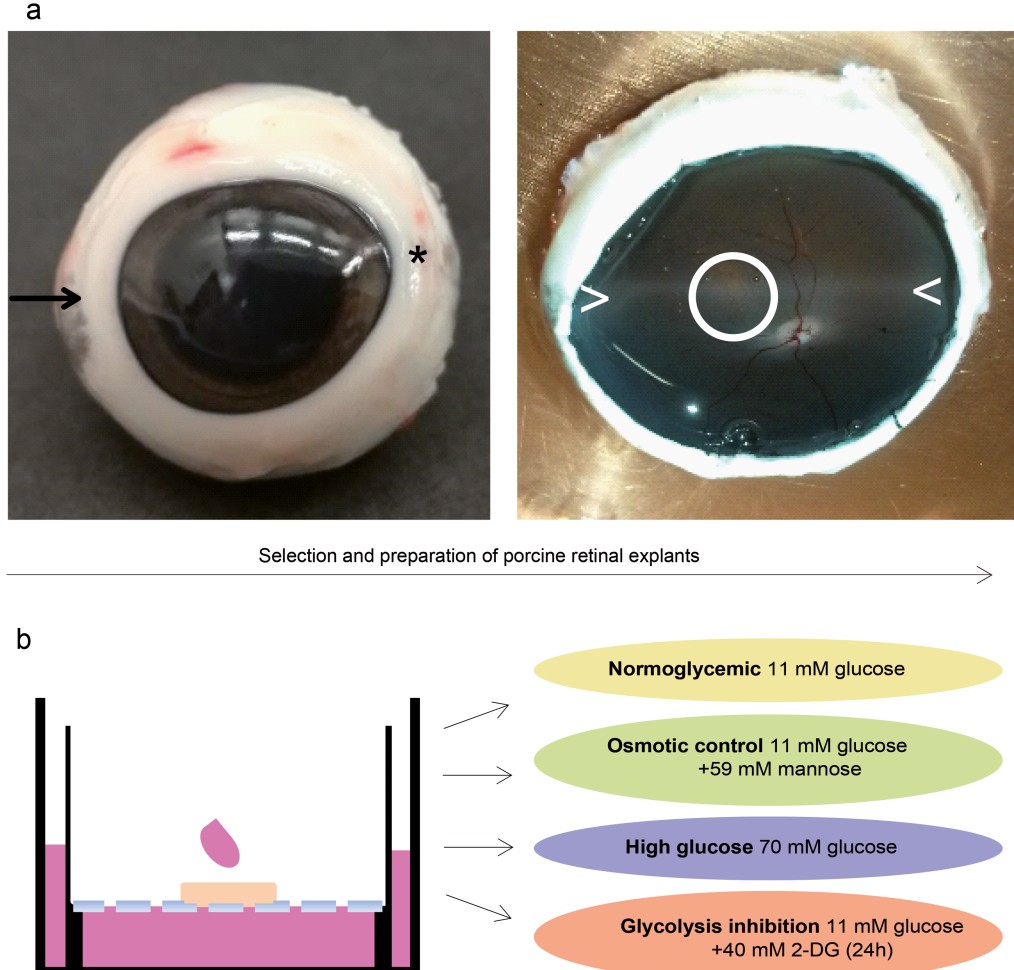

Selection and preparation of porcine retinal explants

**Normoglycemic** 11 mM glucose

**Osmotic control** 11 mM glucose +59 mM mannose

**High glucose** 70 mM glucose

**Glycolysis inhibition** 11 mM glucose +40 mM 2-DG (24h)

72h diabetic treatment

**Figure 4  Validation in a porcine diabetic retinal explant culture model.** (A) Left: Nasal pole of a left pig eye, recognizable by the slight bulge of the eyeball (arrow). At the temporal pole, the corneal limbus tapers (asterisk). The optic nerve on the back of the eyeball is located slightly below the horizontal center and slightly temporal. Right: The same eye opened. A horizontal band extending from the nasal to the temporal pole nearby the optic disk (illustrated by ><) is called "visual streak" and is distinguished by high densities of cones and ganglion cells (*Garca et al., 2005*; *Hendrickson & Hicks, 2002*). Explants were excised from this region dorso-nasal of the optic disc (circle). (B) Explants were cultured at the fluid/air interface on polycarbonate membrane with one drop of medium on the top. Treatments were performed for three days.

Further, SPP1 was recently found to be secreted by porcine RMG in vitro, exhibiting a survival-promoting effect on porcine retinal ganglion cells in vitro (*Ruzafa et al., 2018b*).

To analyze SPP1 in the context of the retinal network of RMG cells, we next used porcine retinal explants as a model. We took pieces from the visual streak (Fig. 4A), a region of the porcine retina which is characterized by very high densities of cones and ganglion cells, in a ratio that is also found in humans (*Garca et al., 2005*; *Hendrickson & Hicks, 2002*).

Explants were cultured on the fluid/air interface on a polycarbonate membrane (Fig. 4B). In reference to the isolated porcine RMG, short term diabetic treatment was applied. An osmotic control group was included to determine the osmotic effect independently of the effect of high glucose or glycolysis inhibition to the explants (Fig. 4B).

Since SPP1 expression patterns in porcine retina were not reported so far to our knowledge, we initially examined it's distribution by immunohistochemical fluorescence staining of porcine retinal organotypic explant cultures after three days in vitro under normoglycemic conditions (Fig. 5).

SPP1 was detected at the level of photoreceptor outer segments (POS) and in the inner neuroretina in the area of the ganglion cell layer (GCL) and nerve fiber layer (NFL) (Fig. 5, top, left panel). Vimentin was used as marker for RMG and its characteristic RMG morphology (Fig. 5, top, mid panel). Double labeling of SPP1 with vimentin revealed co-localization along RMG processes and endfeet in the inner neuroretina (Fig. 5, top, right panel). This co-localization in the area of the GCL and NFL suggests RMG-associated SPP1 expression also in the ex vivo retinal system.

We then aimed to investigate if SPP1 expression in the ex vivo retinal system is downregulated in response to diabetic culture conditions. In all four conditions, vimentin was characteristically expressed in RMG processes throughout the inner retina from the endfeet that build the inner limiting membrane to the outer plexiform layer (Fig. 5, mid panels, vimentin red). In osmotic control and in both diabetic conditions (high glucose and glycolysis inhibitions), the RMG-associated SPP1 diminished in comparison to the normoglycemic cultured explants (Fig. 5, left panels, SPP1 green; overlay with vimentin staining: right panels). Interestingly, SPP1 staining at the level of photoreceptor outer segments remained unaltered in all conditions, indicating a specific RMG response.

## DISCUSSION

Diabetes mellitus is currently one of the major health concerns with nearly half a billion people affected worldwide and a permanently increasing prevalence (*Saeedi et al., 2019*). Chronic hyperglycemia and its consequences are the main issues of diabetes mellitus (*Renner et al., 2013*). Diabetic retinopathy as a microvascular complication is also one of the fastest growing diseases worldwide, leading to blindness and greatly impairs quality of life in affected patients (*Harding et al., 2019*). To date, the mechanisms leading to retinal damage are not well understood.

Appropriate animal models to address the pathogenesis of diabetic retinopathy are scarce to date, because traditional animal models in rodents lack translational value because of significant differences in physiological, anatomical and immunological aspects. The pig recently evolved as a favorable model for diabetes given the marked similarity with the humans with respect to the developed pathophysiological features (*Cole & Florez, 2020*; *Kleinwort et al., 2017*; *Renner et al., 2020*; *Renner et al., 2013*). RMG is the most important glial cell in the retina where it plays a central role (*Bringmann et al., 2006*; *Bringmann & Wiedemann, 2012*) in the pathology of DR. However, the molecular mechanisms of pathological changes to and initiated by this specific cell type are not well understood

(*McDowell et al., 2018*). Recently, a proteomic study in diabetic patients demonstrated retinol binding protein-3 to control glucose transport in RMG and thus development of DR (*Yokomizo et al., 2019*). This study highlights not only the successful application of proteomic techniques to elucidate molecular pathomechanisms in DR, but also the important role of RMG in pathologic processes of diabetic retina. To the best of our knowledge proteome-wide responses of RMG to diabetes-like metabolic situations have not been studied before and by using a comprehensive, differential proteomic analysis in our primary RMG cell culture model (*Hauck, Suppmann & Ueffing, 2003*), we were able to investigate specific mechanisms attributed to this particular cell type. Short term treatment in cell culture allowed to shed light on the mechanisms underlying the initial phases of disturbed glucose metabolism. Applying a differential proteome analysis approach, we observed highly significant proteome changes in cells cultured under two different treatment regimens designed to mimic fluctuating glucose levels naturally occurring in diabetes. A total of 65 proteins showed a significant differential expression upon both diabetes-like conditions. A pathway enrichment analysis of the proteins with altered abundance revealed an enrichment of ECM associated pathways, with a hinted downregulation of ECM organization and cell adhesion. Decreased production of ECM proteins under diabetic conditions could explain our previous finding of the disruption of the inner limiting membrane (ILM) in a diabetic pig model resulting in accumulation of hard exudates and cotton-wool spots, which are common features of proliferative neovascularization (*Kleinwort et al., 2017*).

The ILM is a thin sheet composed of ECM proteins at the inner surface of the retina closely connected to RMG endfeet (*Syrbe et al., 2018*). There is evidence that human RMG produce ECM proteins suggesting their role in the ILM assembly (*Ponsioen et al., 2008*). The present data point toward a remodeling of ECM proteins which potentially destabilizes the ILM thereby impairing its integrity in DR. Further support to our hypothesis on RMG involvement in de-stabilizing ILM is provided by the observed downregulation of integrins, which are key factors for the correct assembly and function of ILM (*Halfter et al., 2008*).

Since the retina comprises a highly sophisticated neuronal network of many specialized cells in close interaction, we aimed to confirm our findings in a model which closely resembles in vivo conditions. Hence we established porcine organotypic retinal explants cultures which preserve the cellular architecture and functional interactions physiologically present in the tissue hence providing an adequate and comprehensive model of the retina (*Schnichels et al., 2019*).

In this study, porcine retinal explants were excised from the visual streak, a region of the porcine retina which is characterized by high densities of cones and ganglion cells (*Garca et al., 2005*; *Hendrickson & Hicks, 2002*). The region dorso-nasal the optic disc within the visual streak contains the highest cone numbers and thus resembles the human macula (*Hendrickson & Hicks, 2002*; *Kleinwort et al., 2017*). We therefore examined a retinal region with a similar cellular distribution like the human macula to use the advantage of the porcine model to learn about human macula diseases.

We identified SPP1 (synonym osteopontin) as a major changed candidate in our study. SPP1 is an interesting molecule with pleiotropic functions (*Ruzafa et al., 2018a*). The

function in the retina is not entirely clear to date (*Ruzafa et al., 2018a*). SPP1 influences cell survival and attachment, cell migration, inflammation, migration and homeostasis after injury (*Ruzafa et al., 2018a*). In this study, SPP1 disappeared significantly and rapidly in short term hyperglycemic conditions in cell culture models and likewise in organotypic cultures. Although the exact implications for diabetic retinopathy remain open at this point, the loss of a SPP1, which has been described to rescue photoreceptors and to be induced by neuroprotective factor GDNF (*Del Rio et al., 2011*), is an important finding of our study and deserves further investigation in our opinion. Interestingly, the loss of protein abundance was not limited to SPP1, but was also observed for its known interactors, the integrins (*Yang et al., 2019*).

Here, among other integrins, ITGB1 was downregulated under simulated diabetic conditions. Conditional deletion of ITBG1 in mouse brain leads to partial gliosis of astrocytes (*Robel et al., 2009*). Gliosis is generally considered as a protective response of glial cells to harmful insults (*Bringmann et al., 2006*; *Burda & Sofroniew, 2014*) and occurs in RMG probably in an early stage of DR (*Vujosevic et al., 2015*). However, in contrast to the observed downregulation of ITGB1, GFAP and vimentin, two prototype markers of gliosis, were not changed by modifying glucose levels in RMG (see Table S1). Therefore, we speculate that changes in ECM proteins related to gliosis could be triggered by high glucose treatment even before upregulation of intermediate filament proteins (*Liu et al., 2016*), as well established marker for reactive RMG gliosis (*Subirada et al., 2018*).

Our results may indicate an involvement of RMG in the neurodegenerative processes in early DR, because SPP1 protein was consistently found less abundant in cultured RMG under diabetic treatment conditions. To date, we do not know the molecular mechanism leading to decrease in SPP1 in these conditions. Potential effects of hyperglycemia in diabetes range from specific metabolic effects, through non-enzymatic glycation, glycoxidation, lipoxidation and osmotic stress (*Yu et al., 2007*). In our cell culture environment, mannose was applied to specifically induce osmotic effects in the explants and to distinguish them from the other hyperglycemia-induced mechanisms (*Yu et al., 2007*). Since the observed effects on the abundance of SPP1 also occurred in the osmose control group, we speculate that alterations in the SPP1 protein levels might result from a multi-causative osmotic stress. This report is the first to describe a RMG specific SPP1 regulation in a high glucose and hyperosmolar state. Since the hyperglycemic hyperosmolar state is a serious acute metabolic complication of diabetes mellitus (*Umpierrez & Korytkowski, 2016*), further experiments should unravel potential relations of hyperglycemic and osmotic conditions that induce decrease in SPP1 levels.

Interestingly, it has been shown that supplemented SPP1 inhibits experimentally induced osmotic swelling in ex vivo rat RMG (*Wahl et al., 2013*). Glial swelling is involved in pathology of diabetic macula edema, a severe complication of DR (*Coughlin, Feenstra & Mohr, 2017*; *Daruich et al., 2018*; *Graue-Hernandez et al., 2020*; *Wahl et al., 2013*). We speculate that a decrease in SPP1 expression level might contribute to RMG swelling in diabetic macular edema. In the retina of diabetic pigs we observed a central swelling of the NFL (*Kleinwort et al., 2017*). It is reasonable to believe that decreased expression of SPP1 could account for the swelling of glial cells as well as the surrounding structures.

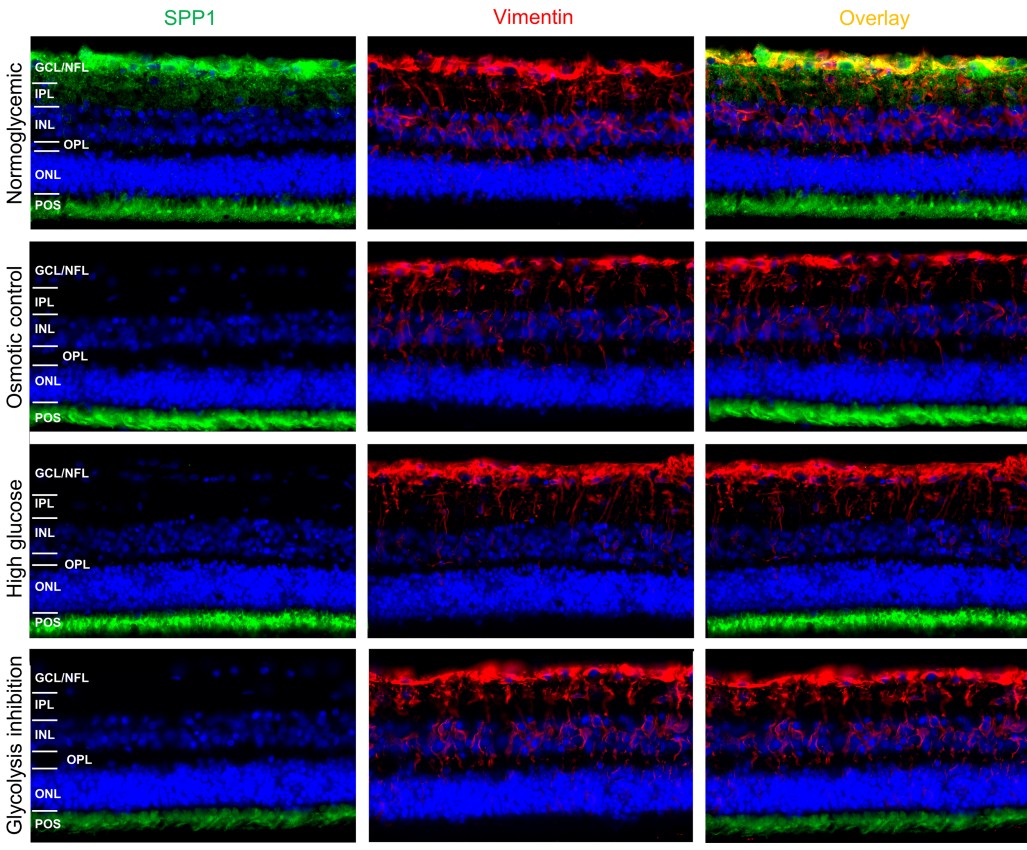

**Figure 5** **Immunohistochemical verification of SPP1 abundances in retinal organotypic explants upon different treatment conditions.** Retinal expression of SPP1 (left panel, green) and vimentin (mid panel, red) in representative explants cultured under normoglycemic ($n = 4$), osmotic control ($n = 4$), high glucose ($n = 4$) and glycolysis inhibition ($n = 4$) conditions for 72 h. SPP1 was expressed at the level of POS independently of the culture condition. Expression along the inner neuroretina in the normoglycemic condition was associated with RMG, as demonstrated by the overlay image of SPP1 and vimentin stainings (top, right panel, yellow), but diminished in all other conditions. Nuclei are stained with HOECHST in blue. GCL, ganglion cell layer; NFL, nerve fiber layer; IPL, inner plexiform layer; INL, inner nuclear layer; OPL, outer plexiform layer; ONL, outer nuclear layer; POS, photoreceptor outer segments.

Noteworthy is the neuroprotective function of SPP1 in retina. In the porcine model, a pro-survival effect of SPP1 on in vitro retinal ganglion cells (*Ruzafa et al., 2018b*) and on photoreceptors (*Del Rio et al., 2011*) has been described. We believe that under diabetic conditions, reduced expression of RMG-associated SPP1 might contribute to severe neuronal damage in DR *in vivo*. Investigating whether neuronal survival in porcine retinal explants is affected by RMG-associated SPP1 expression will be the focus of our future studies. Interestingly, we have also previously found a significant downregulation of RMG-associated SPP1 expression in retinas of horses with uveitis (*Deeg et al., 2011*). Collectively, these data point to SPP1 being generally involved in retinal pathology.

## CONCLUSIONS

Our findings suggest an initial involvement of RMG in DR. Perturbation with glucose levels significantly alters many ECM proteins in RMG suggesting profound ECM remodeling; secreted phosphoprotein 1 (SPP1) being one of these. Given the previously described neuroprotective role of SPP1 and its inhibitory role on RMG cell swelling, SPP1 could hold potential as candidate molecule for developing treatment strategies. Modulating SPP1 levels, as well as other components of the ECM could help to prevent neurodegeneration in DR and RMG might represent promising targets in such circumstances.

## ACKNOWLEDGEMENTS

We would like to thank Linnea Taylor from the Ophthalmology Department, Medical Faculty, Lund University for her help in establishing the porcine retinal explants in our lab and critical discussions.

### Funding

This work was funded by grants from the Deutsche Forschungsgemeinschaft: HA 6014/5-1 (to Stefanie M. Hauck) und DE 719/7-1 (to Cornelia A. Deeg) in the priority program SPP 2127. The funders had no role in study design, data collection and analysis, decision to publish, or preparation of the manuscript.

### Grant Disclosures

The following grant information was disclosed by the authors:
Deutsche Forschungsgemeinschaft: HA 6014/5-1.
und DE 719/7-1.

### Competing Interests

The authors declare there are no competing interests.

### Author Contributions

- Sandra Sagmeister performed the experiments, analyzed the data, prepared figures and/or tables, authored or reviewed drafts of the paper, and approved the final draft.
- Juliane Merl-Pham performed the experiments, analyzed the data, authored or reviewed drafts of the paper, and approved the final draft.
- Agnese Petrera performed the experiments, prepared figures and/or tables, authored or reviewed drafts of the paper, and approved the final draft.
- Cornelia A. Deeg conceived and designed the experiments, analyzed the data, authored or reviewed drafts of the paper, and approved the final draft.
- Stefanie M. Hauck conceived and designed the experiments, performed the experiments, analyzed the data, prepared figures and/or tables, authored or reviewed drafts of the paper, and approved the final draft.

## Animal Ethics

The following information was supplied relating to ethical approvals (i.e., approving body and any reference numbers):

No experimental animals were used in this study, eyes from healthy adult pigs were received fresh from a local abattoir. Collection and use of porcine eyes from the abattoir was approved for purposes of scientific research by the appropriate board of the veterinary inspection office Munich, Germany (permit number: DE 09 162 0008-21).

## Data Availability

Data are available via ProteomeXchange: PXD015879.

https://www.ebi.ac.uk/pride/archive/projects/PXD015879.

## Supplemental Information

Supplemental information for this article can be found online at http://dx.doi.org/10.7717/peerj.11316#supplemental-information.

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
