# Peer review of "High glucose treatment promotes extracellular matrix proteome remodeling in Mller glial cells"

_PeerJ, doi:10.7717/peerj.11316_

## Round 0.1 · original submission · Major Revisions

Thank you again for your submission. While broadly enthusiastic, the reviewers bring up some questions/concerns that should be addressed to improve the clarity of the manuscript and better assess the results.

·

Basic reporting

This is a very well-written and structured manuscript.

Experimental design

Acceptable.

Validity of the findings

Acceptable.

Additional comments

This is a very-well written manuscript describing a proteomic study on pig primary retinal Müller cells (RMCs) or explants cultured under high glucose conditions. The manuscript does provide new information in the field: potential protein alteration pattern in RMCs in diabetic retina and new information about osteopontin under similar conditions. However, following changes may be necessary/helpful:
1. As there are several proteomic studies related to RMCs, the authors should provide a short discussion on the differences (distinctive features) of their results vs that in previous studies.
2. No culture conditions will be completely identical to that in real diabetes. Therefore, an accurate description of experimental procedures/results under diabetes-mimic condition should use something like diabetes-like or high glucose condition. Hyperglycemic is not the correct term.
3. Figure legends should provide sufficient details, eg. Fig 2 legend 1st sentence: “Differential protein abundances in hyperglycemic (a,b) or glycolysis inhibition (c,d) compared to normoglycemic treatment group..” should at least provide info about the source of protein, such as primary cultures, etc.
4. For retinal Müller glial cells, the conventional abbreviation should be RMGCs. If the Müller cell cultures are the experimental focus of this study, the term retinal Müller cells (RMCs) may be a better term to use.
5. Conclusion should add a sentence or two for other major findings, such as significant changes in ECM remodeling—related proteins.

·

Basic reporting

The manuscript is generally well written and presented clearly.

Experimental design

The experimental design and methods to elucidate the hypothesis were adequate.

Validity of the findings

The presentation of the results was clear too. However, some aspects should be revised and the conclusions softened.

Additional comments

The study by Sagmeister et al. examined the effect of short term hyperglycemic as well as glycolysis inhibition on retinal Müller glial cells (MGCs) proteome. The results revealed significant changes in RMG proteome primarily in proteins building the extracellular matrix (ECM) indicating fundamental remodeling processes of ECM as novel rapid response to hyperglycemia. Osteopontin (SPP1) as well as its interacting integrins were significantly downregulated and organotypic retinal explant culture confirmed the selective loss of SPP1 in RMG upon treatment.
The manuscript is generally well written and presented clearly. The experimental design and methods to elucidate the hypothesis were adequate. The presentation of the results was clear too. However, some aspects should be revised and the conclusions softened.

Major concerns:

1. In lines 65 and 67 the authors refer to diabetes but they do not reference the type. Clarify this important point. In addition, in line 118 they state that the 2-DG uptake competes with glucose, as it is transferred by the same transporters. Define which transporters the authors refer to.

2. Usually, the glucose concentration used in culture of RMG under hypeglycemia conditions is 25 mM. Please provide information and references about the selected concentration of 35 mM of glucosa (line 112). Was this concentration related with DR in humans?

3. In the same way (line 233), authors should provide more detailed information and references related with the hyperglycemic explants treatment. Was this concentration (70 mM) toxic? How was the viability of the explants affected after 72 h of treatment?

4. In line 412, the autors indicated that they confirmed the findingsobtained in RMG in a model that more closely resembling the in vivo conditions. What exact culture medium was used in the porcine neuroretinal organotypic explant cultures? Did the medium contain hormones, growth factors etc? Usually, the culture medium used is Neurobasal medium (Kuehn 2016, 2017). Please, clarify this important point.

5. According to Pereiro et al., (2020) primary porcine RMG cultures reach confluence after 7 days in vitro. This time point was stablished as the best, since it has been previously demonstrated a possible transdifferentiation of Müller cells after 14 days in vitro (Guidry, 1996; Hauck et al., 2003). How do the authors justify the completion of their studies after the 14days?

6. Due to the neuroprotective function attributed to SPP1, it would be interesting to determine the cell death in retinal explants. With this experiment, the conclusion of the manuscript and abstract will be justified. On the contrary, this paragraph related to SPP1 is overstated and need to be moderated. Please consider this when re-writing this sentence and revisiting your abstract and conclusions.
In addition, in Figure 5, could the authors add a Western blot assay with quantification of RMG under hyperglycemia or glycolisis inhibition also showing the reduction of SPP1 levels? This assay would help to complete the results.

7. In line 437, the authors refer to another protein, ITGB1, which was downregulated under diabetic conditions and its posible effect on the gliosis. In this sense, the authors should include the complementary analysis of GFAP in retinal organotypic explants, which be helpful to add more evidence to the discusión of this manuscript.

Minor concerns:

Line 46: authors should check for punctuation marks

Figure 1 Legend: improve the writing of this paragraph. Pay special attention to… after lysis, cells were lysed and …

---

## Round 0.2 · accepted · Accept

Thank you for addressing the reviewers' concerns and congratulations again.

·

Basic reporting

This is a very well-written and structured manuscript.

Experimental design

Acceptable.

Validity of the findings

Acceptable.

Additional comments

The authors have a dressed most of my concerns. The remaining part(s) can be considered as difference in opinion that I have no objection.

·

Basic reporting

This is a very well-written and structured manuscript.

Experimental design

Acceptable

Validity of the findings

Acceptable

Additional comments

No more comments